# Genome assembly of wisent (*Bison bonasus*) uncovers a deletion that likely inactivates the *THRSP* gene
Chiara Bortoluzzi[1], Xena Marie Mapel[1], Stefan Neuenschwander[2], Fredi Janett[3], Hubert Pausch [1,4] & Alexander S. Leonard [1,4]

The wisent (*Bison bonasus*) is Europe's largest land mammal. We produced a HiFi read-based wisent assembly with a contig N50 value of 91 Mb containing 99.7% of the highly conserved single copy mammalian genes which improves contiguity a thousand-fold over an existing assembly. Extended runs of homozygosity in the wisent genome compromised the separation of the HiFi reads into parental-specific read sets, which resulted in inferior haplotype assemblies. A bovine super-pangenome built with assemblies from wisent, bison, gaur, yak, taurine and indicine cattle identified a 1580 bp deletion removing the protein-coding sequence of *THRSP* encoding thyroid hormone-responsive protein from the wisent and bison genomes. Analysis of 725 sequenced samples across the Bovinae subfamily showed that the deletion is fixed in both *Bison* species but absent in *Bos* and *Bubalus*. The *THRSP* transcript is abundant in adipose, fat, liver, muscle, and mammary gland tissue of *Bos* and *Bubalus*, but absent in bison. This indicates that the deletion likely inactivates *THRSP* in bison. We show that super-pangenomes can reveal potentially trait-associated variation across phylogenies, but also demonstrate that haplotype assemblies from species that went through population bottlenecks warrant scrutiny, as they may have accumulated long runs of homozygosity that complicate phasing.

The wisent (*Bison bonasus*), also known as the European bison, is a member of the *Bovidae* family that contains several domesticated livestock species, such as cattle, buffalo, yak, sheep, and goat[1]. The wisent went extinct in the wild in 1921. A restoration program established in 1942 from 12 captive individuals by the International Society for the Preservation of the European Bison led to the creation of the lowland and lowland-Caucasian lines, which are still managed as separate populations[2,3]. As a result of these efforts, the wild wisent population has expanded to around 6800 free-roaming individuals across 10 countries[4]. However, the wisent is still considered a near threatened species according to the International Union for the Conservation of Nature[5].

Wisent genetic research had long relied on mitochondrial genome sequences and microsatellite markers[3,6,7]. More recent investigations into genome-wide genetic diversity in wisent have utilized a taurine cattle (*Bos taurus taurus*) reference genome and bovine microarrays[8–10]. Considering that wisent and taurine cattle diverged between 1.7 and 0.85 million years ago (MYA)[9], this methodological approach likely introduces reference

bias[11,12]. Low contiguity (contig N50: 14.53 Kb; scaffold N50: 4.69 Mb) and high fragmentation (29,074 scaffolds) complicate the wider application of a short read-based wisent assembly[13] for genetic investigations[14], therefore assembling the wisent genome with more recent approaches is warranted. A contiguous genome assembly of the wisent is also relevant for the Bovine Pangenome Consortium[15] which aims to investigate signatures of domestication, natural and artificial selection in the genomes of divergent lineages of the *Bovinae* subfamily.

The semi-automated construction of highly contiguous, near complete, and near error-free assemblies is feasible with current sequencing and assembly methods[16]. Trio binning is a widely used assembly method when parent-offspring trios are accessible[17–21]. The availability of parental sequencing data allows to bin the offspring's sequencing reads into haplotype-specific sets based on *k*-mers specific to either the paternal or maternal haplotype, thereby assembling maternal and paternal haplotypes[17,18]. Here, we assemble a wisent genome with Pacific Biosciences (PacBio) HiFi reads. We produce two haplotype assemblies with trio binning and a

[1]Animal Genomics, ETH Zurich, Zurich, Switzerland. [2]Animal Genetics, ETH Zurich, Zurich, Switzerland. [3]Clinic of Reproductive Medicine, University of Zurich, Zurich, Switzerland. [4]These authors contributed equally: Hubert Pausch, Alexander S. Leonard. ✉e-mail: hubert.pausch@usys.ethz.ch; alexander.leonard@usys.ethz.ch

primary assembly. We show that the wisent genome contains many extensive runs of homozygosity (ROH), which complicates phasing and results in incomplete haplotype-resolved assemblies. We integrate the primary wisent assembly into a bovine super-pangenome and identify a putatively trait-associated deletion of the entire protein-coding sequence of *THRSP* encoding thyroid hormone-responsive protein which is fixed in wisent and bison.

## Results and discussion
### Assembly of the wisent genome
A draft reference of the wisent genome was previously assembled with short sequencing reads[13] but the assembly is highly fragmented and therefore not suitable for detailed pangenome analyses[22]. To make the wisent genome amenable to pangenome analyses, such as identifying structural variants differing between wisent and other wild and domesticated members of the *Bovinae* subfamily, we sampled tissue from a parent-offspring trio to assemble a wisent genome with long reads through trio binning. We collected 131.1 Gb (or approximately 44x coverage) of HiFi reads, with a mean read length of 19.3 Kb and a mean quality value of 32.4 from a male wisent (hereafter referred to as F1) from a captive population. We further collected approximately 40x, 38x, and 36x coverage of Illumina short reads from the F1, his sire, and dam, respectively. We used the parental short reads to trio-bin the long reads, assigning an unknown/paternal/maternal tag to each read. Overall, the "binnability" of the wisent sample reads was 61.9%, which is low compared to the 79.9%, 85.0%, or 99.2% binnability observed for intra-breed (Braunvieh x Braunvieh)[23], inter-breed (Rätisches Grauvieh x Simmental)[24], or inter-subspecies (Nellore [indicine] x Brown Swiss [taurine])[19] HiFi-sequenced *Bos taurus* crosses (Fig. 1a–d). The F1 sample also had higher variability in binnability along

the genome than observed previously in other bovines, suggesting that in this wisent trio there is less parental-specific sequence used to assign haplotypes, but also a more uneven distribution of such sequence (Fig. 1e; Supplementary Fig. 1).

Regions with high proportions of unassigned reads can safely be treated as homozygous sequences present in both offspring haplotypes. However, regions with moderate levels of unassigned reads and high levels of reads assigned to a single haplotype can introduce an assembly gap in the unrepresented haplotype and likely lead to assembly errors - due to mixing of haplotypes - in the overrepresented haplotype. There was a dam/sire-assigned read bias greater than 5-fold for 213,100-Kb bins in the F1 sample, over three times more than observed in a previously analyzed *Bos taurus taurus* intra-breed (Braunvieh x Braunvieh) F1 sample with only 70 bins with large imbalances (Fig. 1f).

Given the low binnability and biased assignment of parental reads in the F1 sample, we used hifiasm to generate both a primary (collapsed) assembly and two haplotype-resolved assemblies, to examine the impact of assembler phasing assumptions. We calculated contiguity, correctness, and completeness for all three new assemblies, as well as the existing short-read based draft wisent reference genome[13]. Most metrics demonstrate the outstanding quality of our assemblies, with the primary assembly improving contiguity a thousand-fold over the existing draft wisent assembly (Table 1). As a result, both haplotypes and the primary assembly captured a higher fraction of unassembled genome, for an additional 0.3 Gb and 0.5 Gb, respectively, while reducing the number of missing (N) bases from 125 Mb in the draft to 0.05 Mb in the primary assembly. The overwhelming majority (93%; 2.86 Gb) of the primary assembly sequence was in scaffolds that aligned to the 29 autosomes and two sex chromosomes of the *Bos taurus taurus* reference sequence.

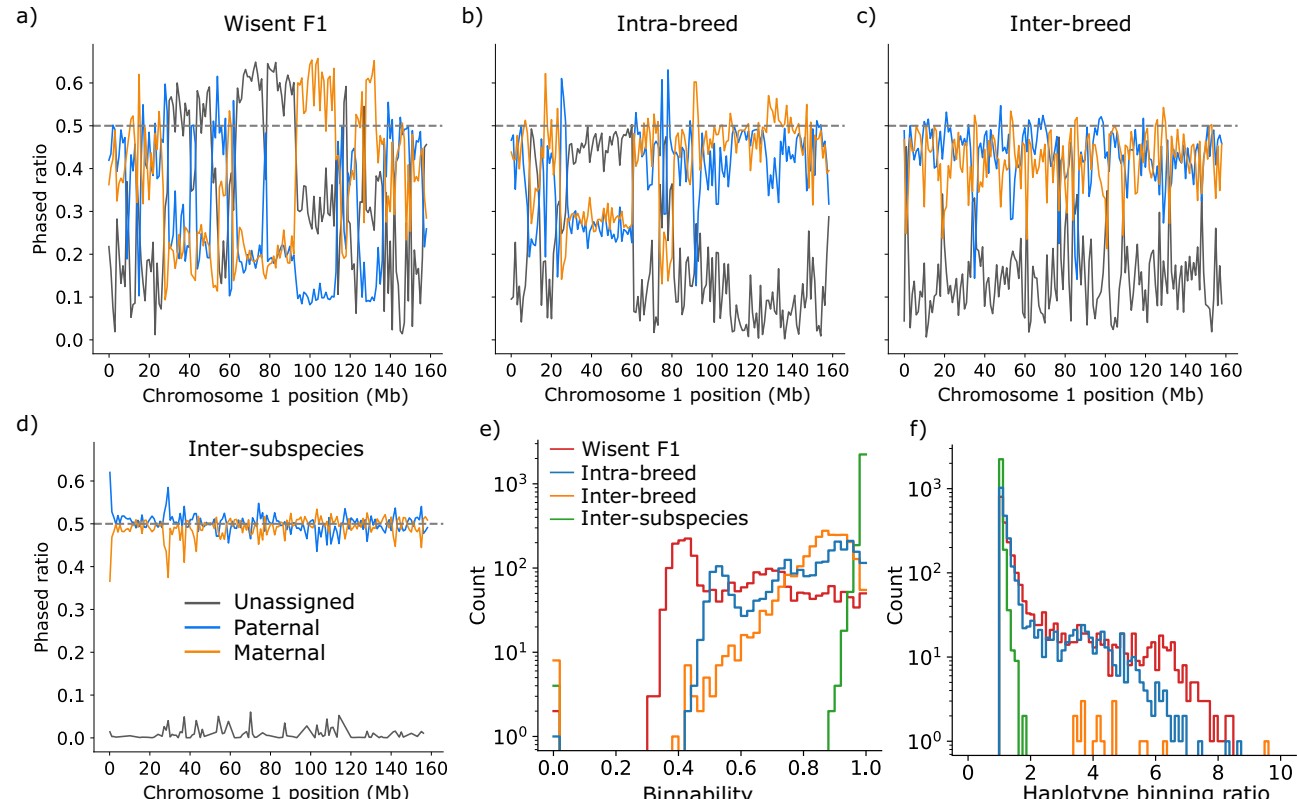

**Fig. 1 | Trio binning of the wisent (Bison bonasus) genome. a–d** Fraction of reads per 100 Kb window tagged as unassigned, paternal, or maternal haplotypes across chromosome 1 for the wisent sample, an intra-breed sample (BSWxBSW), and inter-breed sample (RGVxSIM), and an inter-subspecies sample (NELxBSW). **e** Histogram of proportion of parental-assigned reads (paternal + maternal)/total)

per window across all autosomes. Proportions close to 1 indicate unambiguous phasing. **f** Histogram of paternal assigned read ratio (paternal/maternal if paternal > maternal else maternal/paternal) per window across all autosomes. Values close to 1 indicate balanced phasing, while higher values indicate either paternal or maternal reads are disproportionately assigned. Colors are taken from (**e**).

Table 1 | Assembly statistics of the wisent draft assembly[13], the two haplotype-phased assemblies, and the primary assembly generated in this study

| | | Draft | Haplotype 1 (paternal) | Haplotype 2 (maternal) | Primary |
|---|---|---|---|---|---|
| Assembly accession | | (Wang et al. 2017)[13] | This study | This study | GCA_963879515.1 (this study) |
| Assembled genome size (Gb) | | 2.57 | 2.83 | 2.86 | 3.07 |
| Number of contigs | | 243,242 | 486 | 480 | 248 |
| Contig N50 (Mb) | | 0.02 | 74 | 59 | 91 |
| Number of scaffolds | | 29,074 | 402 | 338 | 210 |
| Scaffold N50 (Mb) | | 4.0 | 95 | 105 | 105 |
| Quality Value | | - | 55.6 | 55.7 | 56.4 |
| Largest scaffold (Mb) | | 31.6 | 164.4 | 164.1 | 163.9 |
| Compleasm (%) | Single copy | 98.4 | 93.4 | 88.4 | 98.5 |
| | Duplicated | 0.6 | 1.1 | 1.1 | 1.2 |
| | Fragmented | 0.7 | 0.3 | 0.3 | 0.2 |
| | Missing | 0.3 | 5.3 | 10.2 | 0.1 |

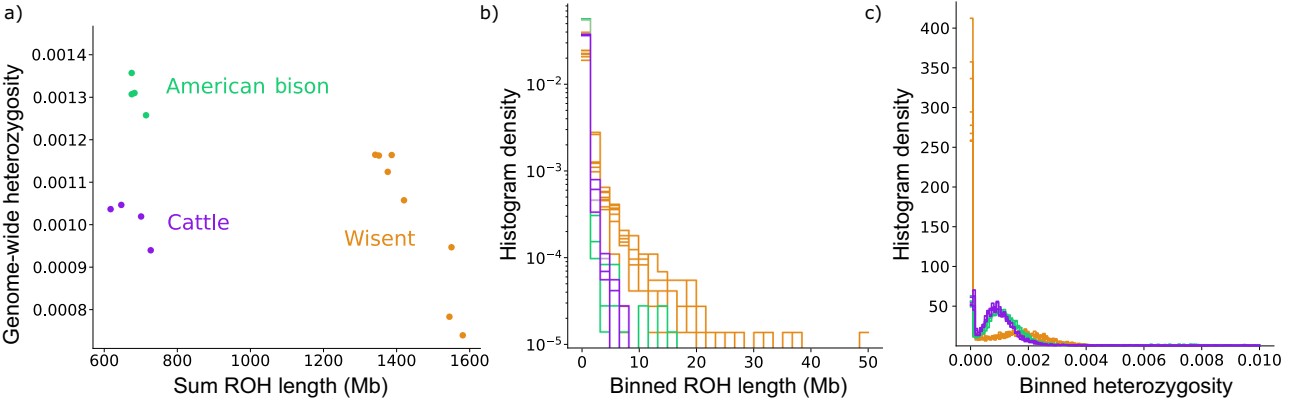

**Fig. 2 | Genome-wide heterozygosity and runs of homozygosity. a** Total length of runs of homozygosity (ROH) versus genome-wide heterozygosity in each wisent ($n = 8$), American bison ($n = 4$), and taurine cattle ($n = 4$) sample. **b** Histogram of ROH lengths derived from 10 Kb bins for the same samples from (**a**), with substantially longer ROH present in wisent. **c** Histogram of heterozygosity in 1 Mb windows for the same samples from (**a**), showing wisent have both many more low heterozygosity bins but also an increase in heterozygosity in some regions.

Assembly completeness was estimated using compleasm and the set of 9226 highly conserved mammalian genes (hereafter called BUSCO genes). The draft and primary assembly had high completeness scores (99.0% vs 99.7%). While the contiguity and correctness of the haplotype-resolved assemblies were comparable to the primary assembly, the gene completeness was noticeably lower, especially for the maternal haplotype at 89.5%. Many of the BUSCO genes missing from one haplotype were found in either the other haplotype or the primary assembly, suggesting that the diploid sequence was incorrectly assigned to only a single haplotype. We confirmed the sporadically missing BUSCO genes should have been in large regions (>Mb) of missing syntenic sequence (Supplementary Fig. 2), rather than local mis-assemblies disrupting BUSCO identification.

**Extended runs of homozygosity complicate haplotype phasing**

The differences in sequence and gene content observed between the haplotype assemblies and the primary assembly were surprising, as we did not encounter such a pattern in other bovine assemblies constructed earlier through trio binning[19] and so prompted a detailed investigation. We looked at the distribution and location of regions with lower-than-expected heterozygosity (runs of homozygosity – ROH) by aligning short-reads of the F1 to the primary wisent assembly to examine whether excessive ROH had an impact on the assembly construction. Variants called from these alignments revealed that ROH covered a total of 1.38 Gb in the F1 genome corresponding to a genomic inbreeding coefficient ($F_{ROH}$) of 0.52 (Fig. 2a;

Supplementary Table 1). Of these, 139 were longer than 2 Mb and tended to correspond with large regions of missing sequence in the haplotype assemblies, which appeared to derive from the hifiasm unitig graph, where unbalanced assignment of paternal phases lead to incorrect haplotype separation (Supplementary Fig. 2). Although many of the ROH were correctly assembled, it was striking to observe such a correspondence between regions of lower-than-expected heterozygosity and regions that are difficult to resolve in the assembly using a trio binning approach. Given the high binning variability of the wisent F1, using a bin-first-then-assemble method[18] encounters similar haplotype-resolved assembly issues (Supplementary Table 2), partially improving BUSCO score (although some previously missing loci are still missing, including the region from Supplementary Fig. 2), but considerably worsening the haplotype switch rate and contiguity.

The genomic inbreeding of the F1 was exceptionally high compared to what is commonly observed in other bovines but was consistent with what we observed in other wisent samples. ROH covered a large fraction of the genome also in the seven (5 captive, 2 wild) additional wisents we investigated (Fig. 2a; Supplementary Table 1). On average, 1.44 Gb of their genomes were in ROH resulting in a $F_{ROH}$ of 0.56. Between 121 and 168 ROH longer than 2 Mb were observed in the eight wisents studied. These long homozygous regions likely reflect founder effects resulting from the genetic bottleneck and expansion of the wisent population at the beginning of the 20th century[9]. Compared to the wisent samples, we found fewer ROH in

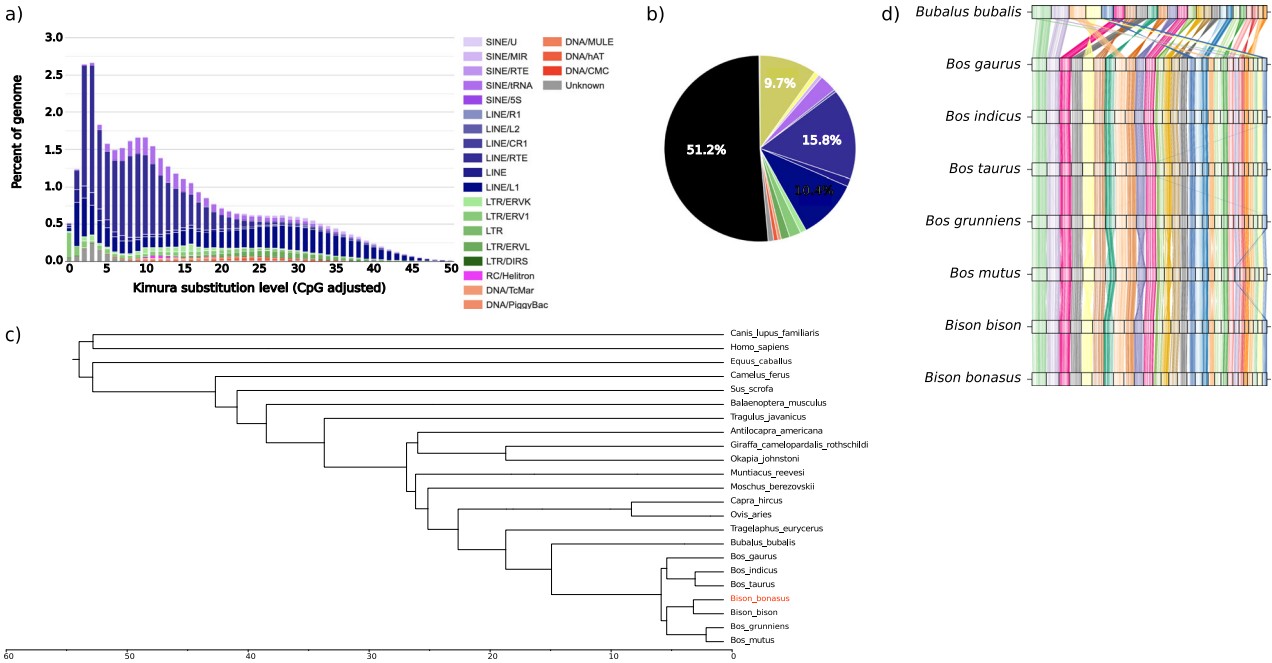

**Fig. 3 | Primary assembly of the wisent (*Bison bonasus*). a** Kimura substitution levels between the repeat consensus and its copies. The histogram plot shows the age distribution of transposable elements (TEs). The total amount of DNA in each TE class was split into bins of 5% Kimura divergence. **b** Pie chart representing the percentage of the genome in different repeats. Repeats are colored following the legend in (**a**), whereas the % of unmasked genome is in black. **c** Phylogenetic tree constructed from single copy BUSCO genes identified in compleasm. The axis shows the divergence time in million years ago (MYA). All nodes were supported in more than 94% of bootstrapped iterations. **d** Synteny plot showing the conservation of large-scale gene linkage and gene order across 8 species. The conserved unique single copy BUSCO genes are connected by lines according to their chromosomal location. Chromosomes are ordered by total size, from the largest to the smallest. Sex chromosomes and the mitochondrial genome are excluded from this analysis.

American bisons and taurine cattle covering only a quarter of their genomes (American bison: 662 Mb; taurine cattle: 665 Mb) (Fig. 2a). ROH longer than 2 Mb were eight- and two-fold less abundant in American bison and cattle, respectively (Fig. 2b). A relatively low $F_{ROH}$ value in American bison is likely the result of hybridization with various domestic cattle breeds, which has been encouraged since the late 1800's, when most of the surviving bison individuals were maintained by cattle ranchers in private herds[25].

Genome-wide heterozygosity in the wisent ($1.04 \pm 0.16 \times 10^{-3}$) was similar to that of cattle ($1.04 \pm 0.22 \times 10^{-3}$) but lower than in American bison ($1.35 \pm 0.19 \times 10^{-3}$), which is in line with previous studies[8,9]. Heterozygosity varied strongly along the genome in wisent as evidenced by an excess of regions with almost no heterozygous sites (Fig. 2c). However, some genomic regions had higher heterozygosity in wisent than in cattle or American bison, reflecting their larger ancient effective population size[9]. These findings corroborate that genome-wide heterozygosity levels are unable to reflect demographic effects that can lead to extended segments of homozygosity and are therefore of limited utility to assess variability of populations[26]. Moreover, our findings emphasize that the average genome-wide heterozygosity can be a misleading metric to consider when conducting haplotype-resolved analyses, as long stretches of the genome may be homozygous and thus unable to be assigned into haplotypes, even for relatively "normal" genome-wide heterozygosity levels.

**Repeat and gene content of the wisent assembly**

The high accuracy of HiFi reads can benefit the assembly of repetitive sequence[27], and so we investigated the repeat content in the new wisent assembly. Since all examined metrics provide confidence that the primary wisent assembly is highly contiguous and near complete, we used it for all downstream analyses. Repetitive elements were identified and classified using a wisent-specific de novo repeat library constructed with Repeat-Modeler. This approach showed that 48.90% of the wisent genome are repetitive sequence, which was similar to that of the draft assembly (47.30%),

American bison (43.52%), and domestic taurine cattle (41.73%) when equally using species-specific de novo repeat libraries (Supplementary Table 3). Within the first class of transposable elements (TEs), also called retrotransposons, long interspersed nuclear elements (LINEs) were the most abundant (27.76%) type, which is also in agreement with their high prevalence in the bovine genome[28], followed by long terminal repeats (LTRs, 4.44%) and short interspersed nuclear elements (SINEs, 3.82%). We found that substantially more satellite DNA was in the HiFi-based wisent (9.69%) than in the American bison (2.04%) and cattle (0.07%) assemblies. Unclassified repeats made up just 0.85% of the wisent repeat content. While this value was similar to that of the American bison (0.88%), it was 11-fold larger than in domestic cattle (0.08%). We ran another round of repeat masking on the American bison and cattle genome using the wisent-specific repeat library generated with RepeatModeler to test the impact of the repeat library on the identification and classification of repetitive sequence. We found that the overall repeat content did not change substantially when using either a species-specific (American bison: 43.52%; cattle: 41.73%) or a wisent-specific repeat library (American bison: 43.27%; cattle: 41.90%), suggesting that de novo repeat libraries are not sensitive enough to identify de novo repeats in highly similar genomes (1% divergence between wisent and bison and wisent and cattle) due to the levels of noise (Supplementary Table 3).

We performed a Kimura distance-based copy divergence analysis of TEs in the wisent assembly to estimate the age of TEs. We observed a predominance of young LINEs and LTRs, as evident from their clustering on the left side of the graph, which indicates minimal deviation from the consensus sequence (Fig. 3a). Additionally, the wisent assembly contained unidentified young repeat copies, classified as "unknown" in the graph. LINEs and LTRs were also the most abundant type of ancient or degenerated TEs, as indicated by their clustering on the right side of the graph.

We used BUSCO genes shared between the wisent and 22 other species to build a phylogenetic tree, which was then used to estimate divergence

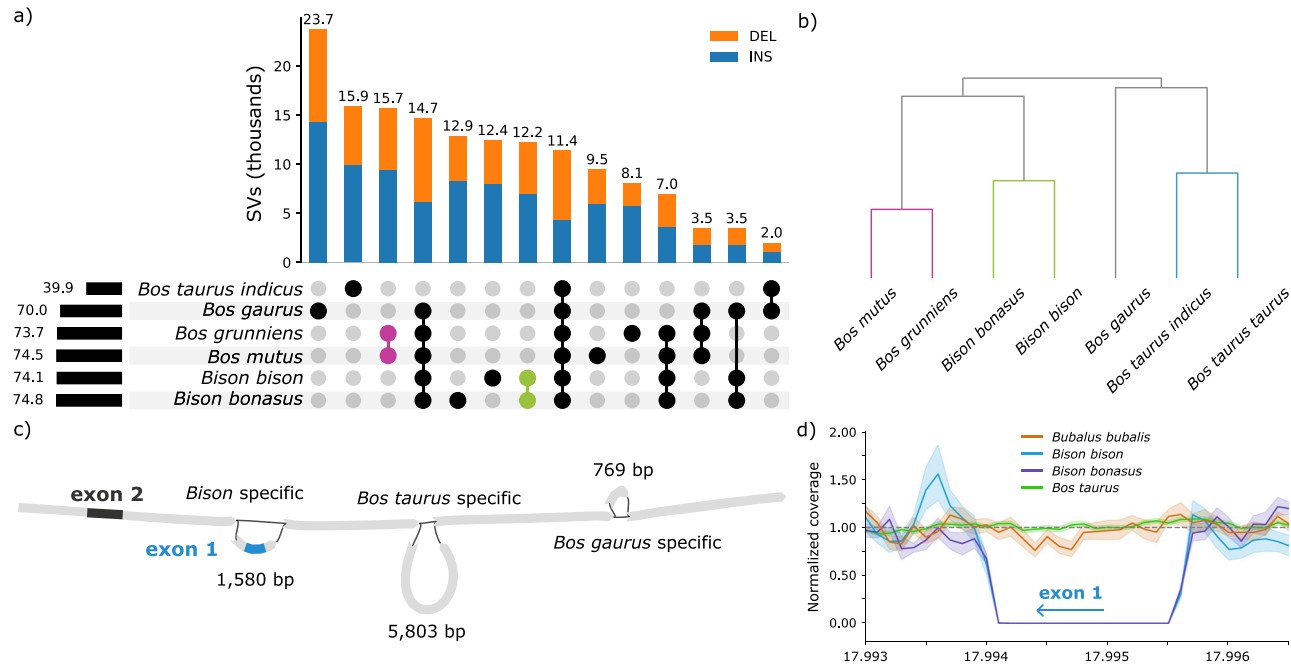

**Fig. 4 | Structural variant analysis from a seven-assembly super-pangenome.**
**a** UpSet plot of SVs called from the 29 autosomes, where the total number of SVs per assembly is shown on the left and the number of intersecting SVs shown above each grouping. The pink and green markers highlight SVs private to both yaks or bison. **b** Relationship tree inferred from pairwise overlaps of SVs. Assemblies are correctly grouped into yak (purple), bison (green), and cattle (blue) clades. **c** A 1580 bp deletion that includes the first coding exon of the THRSP gene was detected in the wisent and bison assemblies. **d** Normalized coverage of biologically independent short read-sequencing data of wisent ($n = 20$), bison ($n = 19$), cattle ($n = 674$), and water buffalo ($n = 12$) samples with at least 5-fold coverage around the 1580 bp deletion. Error bars represent the 95% confidence intervals, and the dashed line indicates the expected normalized coverage of 1.

times (Fig. 3c). The topology and divergence estimates of the *Bovini* subset of the tree are generally consistent with a phylogenetic reconstruction from nuclear whole-genome sequences. Gaur has been previously reported to be more closely related to bison and yak than to cattle[29–31], although a slightly lower bootstrap confidence (Supplementary Fig. 3) and only using a single representative assembly for each external node may explain this minor discrepancy. The wisent formed a clade with the American bison after diverging from this species approximately 2 MYA. This clade was sister to the domestic (*Bos grunniens*) and wild yak (*Bos mutus*) clade, from which it diverged approximately 5 MYA. The domestic taurine (*Bos taurus taurus*) and indicine (*Bos taurus indicus*) cattle formed a clade of their own, with the gaur (*Bos gaurus*) acting as an outgroup. As expected, the water buffalo (*Bubalus bubalis*) was the most distantly related species within the sub-family, diverging from the other *Bos* and *Bison* species approximately 15 MYA. BUSCO genes were also used to perform a synteny analysis to investigate how gene order has changed during the evolution of species within the *Bovinae* subfamily. We found gene order to be conserved between wisent, bison and various *Bos* species but, as expected, gene order is less conserved when compared to distantly related species, such as the water buffalo (Fig. 3d).

**Pangenome analysis reveals a *Bison*-specific deletion inactivating *THRSP***

We then built per-autosome super-pangenomes with the five *Bos* and two *Bison* assemblies, excluding *Bubalus bubalis* due to the different assembled karyotype. Although *Bos gaurus* has a Robertsonian translocation between chromosomes 1 and 29[32], this genome was not assembled through the centromere fusion, leaving 29 separate assembled autosomes and so was included. We assessed the structural variant (SV) diversity across the assemblies (Fig. 4a), finding the wisent sample contains 74,770 SVs (insertions: 37,814, deletions: 36,956) relative to *Bos taurus taurus*, matching previous findings for other distantly related bovids[33,34]. We find many SVs private to either the American bison or wisent, contrasting to wild or

domestic yak which have fewer private SVs, as expected given the more recent split of yak. We find a pronounced increase of SVs private to wisent on chromosome 7 between 10 and 10.6 Mb (27 times higher than the genome-wide rate of private wisent SVs). Many of the private SVs were near or overlapping genes in the olfactory receptor 7 subfamily A (*OR7A*), with a total of 12 annotated protein coding genes in this region (Supplementary Fig. 4), suggesting wisent may have unique variants mediating olfaction compared to the other bovids considered here. Using pairwise overlap of SVs, we can infer a relationship tree (Fig. 4b), closely matching the more rigorously constructed BUSCO gene-based phylogeny discussed previously (Fig. 3c), demonstrating that SVs also reflect evolutionary histories and are a rich source of variation to analyze.

We find 12,217 SVs uniquely common to *Bison* (i.e., American bison and wisent), including 96 that are predicted to have a high impact on proteins (Supplementary Data 1). Among them, a 1580 bp deletion which entirely overlaps the coding sequence of *THRSP* encoding thyroid hormone-responsive protein. *THRSP* has two exons, of which the first contains protein-coding sequence and the second is non-coding. The deletion is predicted to remove the protein-coding exon of *THRSP* in the American bison and wisent haplotypes that were integrated into the super-pangenome (Fig. 4c, Supplementary Fig. 5a). This deletion occurs in the homozygous state in all short read-sequenced bisons ($n = 19$) and wisents ($n = 20$) we investigated, indicating it is likely fixed in both *Bison* species (Fig. 4d, Supplementary Fig. 5b). The 1580 bp deletion was neither detected in the *Bos taurus*, *Bos gaurus*, *Bos mutus* and *Bos grunniens* assemblies that were part of the super-pangenome, nor in any of the short-read sequenced *Bos taurus taurus* and *Bos taurus indicus* ($n = 674$) and *Bubalus bubalis* ($n = 12$) samples. As such, the deletion likely occurred in a common ancestor of bison and wisent, after divergence from the other species of the *Bovinae* subfamily. Genetic drift or selective advantages may have led to the fixation of the deletion in wisent and bison.

We lifted the *THRSP* gene annotation from the *Bos taurus taurus* assembly (ARS-UCD1.2) to the haplotype-resolved, primary, and draft

assemblies of wisent, finding alignment of the non-coding second exon in the paternal haplotype and primary assembly. We confirm that the partial *THRSP* gene is contained within a previously reported synteny group[35] corroborating this region is evolutionarily highly conserved and likely under similar transcriptional regulation and function across species. As expected, given the deletion uncovered from the pangenome, and in line with a partial alignment of *THRSP* in a highly contiguous American bison assembly[36] as well as the missing *THRSP* gene in the highly fragmented American bison reference assembly (GenBank accession: GCA_000754665.1)[37], the coding exon was missing in the primary wisent assembly. The liftover of the coding and non-coding *THRSP* exons was not successful for the previous draft assembly and the maternal haplotype assembly, demonstrating the utility of our near-complete and highly contiguous primary assembly for genomic investigations.

The amino acid sequence of THRSP is evolutionarily highly conserved (Supplementary Fig. 5e). The expression of *THRSP* varies across tissues but it is elevated in tissues that synthesize fatty acids[38]. Comprehensive transcriptomic data ($n = 8642$) from the cattle Genotype-Tissue Expression (GTEx) project[39] confirm that *THRSP* is highly expressed in adipose tissue (5351 TPM), intramuscular fat (66 TPM), liver (34 TPM), muscle (24 TPM) and lactating mammary gland (14 TPM) of cattle (Supplementary Fig. 5). We hypothesized that deletion of the first exon including the entire protein-coding sequence in wisent and bison represents a functional knock-out of *THRSP*. Transcriptomic data are not available for wisent but bison RNA-seq data from a three years old cow are publicly available for liver, spleen, lung, skeletal muscle, kidney and supramammary lymph node tissues[37]. We mapped the bison transcriptomes to the *Bos taurus taurus* reference sequence and compared gene expression with age- and sex-matched bovine samples from cattle GTEx for liver and muscle, i.e., two tissues with high *THRSP* expression and a decent number of informative GTEx samples ($n_{liver}= 14$; $n_{muscle} = 43$). The Spearman correlation coefficient estimated for 17,150 genes was 0.880 and 0.879 for liver and muscle tissue, respectively, indicating that overall gene expression levels in these tissues correlate well between bison and cattle. There were 13 and 8 genes in liver and muscle, respectively, which were highly expressed (>20 TPM) in cattle, but not expressed ($0 \leq TPM < 0.05$) in bison. Only two genes, *THRSP* and *MSMP*, exhibited disparate expression patterns across both tissues, highlighting the rarity of this phenomenon. As expected, given the deletion, we did not detect expression of the coding exon of *THRSP* in any of the bison tissues (Supplementary Fig. 5f). Interestingly, the non-coding second exon was also not expressed in any of the bison tissues, although it is not affected by the deletion. We then aligned 73 transcriptomes from 19 tissues from 4 water buffaloes[40] against the *Bos taurus taurus* reference sequence. Water buffalo is substantially more diverged from cattle than wisent and bison, but the expression profile of both *THRSP* exons was similar to cattle with the highest transcript abundance in adipose tissue (9776 TPM), mammary gland (1570 TPM), skin (440 TPM), liver (298 TPM), and skeletal muscle (37 TPM) (Supplementary Fig. 5f). Collectively, these findings suggest that the non-coding exon does not produce mRNA in bison which supports that the deletion of the coding first exon inactivates *THRSP* and that bison and wisent are lacking the thyroid hormone-responsive protein.

Given the crucial contribution of THRSP in lipogenesis and fatty acid synthesis in the mammary gland and other tissues[38,41,42], we suspect that lack of THRSP impacts lipid metabolism in the two *Bison* species. Mice lacking THRSP produce milk with significantly less medium-chain fatty acids resulting from a decreased lipogenesis in the mammary gland[42]. Neither bison nor wisent have been domesticated, and so the composition of their milk has not been investigated. However, bison meat has lower fat than beef[43,44] which agrees with reduced accumulation of fat in adult *THRSP* knockout mice[45]. While *THRSP* mRNA expression in skeletal muscle tissue is correlated with intramuscular fat content in crossbred cattle[46], the precise function of THRSP in the deposition of intramuscular fat remains to be elucidated[47]. Bison and wisent appear as intriguing model organisms to study the impact of missing THRSP on transcriptional changes in lipid metabolism pathways and to investigate a possible causal relationship

between *THRSP* expression and fat accumulation. Hybridization between *Bison* and domestic cattle which have a functional *THRSP* gene is relatively common[48], and so the phenotypic and genetic diversity of the offspring can be exploited to investigate functional consequences arising from lack of THRSP in natural knockouts.

## Methods

### Ethics statement
No animals were sampled for this study. No ethics approval was required for this study.

### Sample selection
Blood samples of six captive wisents were provided by the Bern animal park and Langenberg animal park in Switzerland, respectively (Supplementary Data 2). Blood samples of five wisents were collected prior to our study for the purpose of establishing a biobank at the animal park. Another blood sample was collected from one wisent after it was killed. The decision to kill the animal was independent from our study. High-molecular weight DNA was extracted from blood using the Qiagen MagAttract HMW DNA Kit.

### Long-read (Pacific Biosciences) and short-read (Illumina) sequencing
A DNA sample from a male wisent (F1) was used for PacBio HiFi sequencing. DNA fragment length and quality were assessed with the Femto Pulse system (Agilent). A HiFi library was prepared and sequenced on three SMRT cells 8 M with a Sequel IIe. We also used an Illumina NovaSeq 6000 machine to generate paired end ($2 \times 150$ bp) reads from DNA extracted from all wisents including the F1 and its parents.

### Genome assembly
The wisent F1 HiFi reads were assembled with hifiasm v0.19.5[17] using default parameters and "-a 5 -n 5 --primary" to produce a primary assembly. We reran hifiasm without the primary flag and instead provided parental *k*-mer databases that were created with yak v0.1 (https://github.com/lh3/yak) from parental short reads to produce two haplotype-resolved assemblies. The contig outputs were then scaffolded to the ARS-UCD2.0 *Bos taurus taurus* reference genome (GenBank assembly accession: GCA_002263795.4) with RagTag v2.1.0[49] primarily to orient and assign chromosome identifiers.

### Assembly validation and completeness
We assessed assembly base-level quality and phasing blocks with merqury (6b5405)[50] using meryl (https://github.com/marbl/meryl) *k*-mer databases created from parental and offspring short read sequencing. We used calN50.js (https://github.com/lh3/calN50) to evaluate the contig contiguity. We ran compleasm v0.2[51] using the *mammalia_odb10* database, which contains 9226 highly conserved genes from 24 species. The hifiasm-processed unitig graph was visualized with bandageNG v2022.9[52].

### Analysis of haplotype specific F1 reads
We used canu v2.2[53] to assign paternal/maternal/unknown haplotypes to each read, using parental *k*-mer databases created from Illumina reads with meryl v1.3. Tagged reads were then aligned with minimap2 v2.26[54] using the map-hifi preset to the ARS-UCD2.0 *Bos taurus taurus* reference genome. The alignments were sorted with SAMtools v1.19.2[55] and the starting coordinate for every primary alignment was recorded for each haplotype tag. We used a 100 Kb window to bin the read counts for each haplotype across the autosomes. We estimated *k*-mer distributions by creating *k*-mer databases and histograms with meryl v1.3 from each individual's HiFi reads. Sample information is given in Supplementary Table 4.

### Repetitive sequence analysis
We generated a wisent-specific de novo repeat library for the primary assembly using RepeatModeler v2.0.4[56] to identify, classify, and mask repetitive elements. RepeatModeler was run in combination with RECON

v1.08[57], RepeatScout v1.0.6[58], and Tandem Repeats Finder v4.09.1[59]. The complete Dfam v3.7[60] and RepBase (final version 10/26/2018)[61] libraries were used to classify repetitive elements based on homology in different repeat families. Consensus sequences obtained from RepeatModeler were used to softmask the genome with RepeatMasker v4.1.5 (https://www.repeatmasker.org/). We ran RepeatMasker by specifying the *rmblast* search engine and the slow search mode. We used the *calcDivergenceFromAlign.pl* script provided by RepeatMasker to summarize the Kimura substitution levels between the repeat consensus and its copies. Repeat landscape plots were produced with the *createRepeatLandscape.pl* script bundled in RepeatMasker. We applied this approach also to the American bison (*Bison bison*) (GenBank assembly accession: GCA_030254855.1) and taurine cattle (*Bos taurus taurus*) (GenBank assembly accession: GCA_002263795.3) assemblies for comparison purposes.

### Phylogenetic tree construction
We constructed a phylogenetic tree from complete single copy orthologs identified by compleasm v0.2[62]. We ran compleasm using the *mammalia_odb10* database on an additional set of 22 species (Supplementary Table 5). We aligned the protein sequence of the 8392 complete single copy genes shared between wisent and 22 other species using MAFFT v7.490[63]. Protein sequence alignments were trimmed using trimAI v1.4[64], specifying a gap threshold of 0.8 and a minimum average similarity of 0.001. Trimmed protein sequences were concatenated to form a supermatrix, which was provided to RaxML v8.2.12[65] to reconstruct a maximum likelihood phylogeny. RaxML was run with the PROTGAMMAJTT model and 1000 bootstrap replicates. Divergence times were estimated in r8s[66] (https://github.com/iTaxoTools/pyr8s) using fossil records previously reported in ref. 1. The resulting phylogenetic tree was visualized in FigTree v1.4.4 (https://github.com/rambaut/figtree).

### Synteny
Synteny was identified using the Chromosomal Orthologous Link analysis approach (https://github.com/chulbioinfo/chrorthlink). We used the set of complete single copy mammalian genes identified with compleasm v0.2 that were used in the phylogenetic tree analysis to assess the conservation of large-scale gene linkage and gene order compared to that of seven other members of the Bovidae family (American bison, wild yak, domestic yak, taurine cattle, indicine cattle, gaur, and water buffalo). Synteny plots were generated in R v4.2.2 using the genoPlotR library[67].

### Annotation of the wisent assemblies
We used liftoff v1.6.3[68] to map the annotation (in GFF) of taurine cattle onto the F1 maternal and paternal haplotypes, the F1 primary assembly, and the existing draft assembly. Liftoff was run using "-copies" to look for extra gene copies in the target genome and "-sc 0.95" to specify a minimum sequence identity in exons/CDS of 95% to consider a gene a copy.

### Structural variants
We constructed per-chromosome pangenomes with minigraph v0.20[69] using "-cxggs -j 0.2" from the five *Bos* and two *Bison* species from the synteny analysis, using the *Bos taurus taurus* reference sequence as backbone and adding assemblies in order of their mash v2.3 divergence[70]. Graph paths (P-lines) were reconstructed using minigraph call, allowing vg v1.55.0 deconstruct[71] to call structural variants (SV) for each assembly using taurine cattle as reference. We used BCFtools query v1.19[72] to print genotypes for each SV, which were then plotted with upsetplot v0.9 (https://github.com/jnothman/UpSetPlot). We estimated the SV-tree considering the reciprocal of number of SVs between each pair of assemblies, followed by applying an UPGMA clustering with SciPy v1.12[73]. The functional impact of SVs was predicted with the Variant Effect Predictor (VEP) tool[74].

### Alignment of short-read DNA samples
We supplemented our short-read dataset with previously generated short-read sequencing data of two male wild wisents (BBO_3569 and BBO_3574)

(BioProject: PRJNA312492)[9]. We also included short-read sequencing data of four American bisons (BioProject: PRJNA343262), and four cattle from European taurine breeds (BioProject: PRJNA176557)[75]. For more information about these samples, refer to Supplementary Data 2. The quality of the raw sequencing data was assessed using the FastQC v0.11.9 software (https://www.bioinformatics.babraham.ac.uk/projects/fastqc/). Reads were aligned to our wisent assembly using the MEM algorithm of the Burrows-Wheeler Alignment (BWA) software v0.7.17-r1188[76] with "-T 20" to output only alignments with mapping quality >20. Duplicate reads were marked with samblaster v0.1.24[77], and SAMtools v1.19.2 was used to convert the SAM file into a binary BAM format. Sambamba v0.8.1[78] was used for coordinate-sorting, and bamtools v2.5.1[79] and Qualimap v2.3[80] were used to assess the quality of the alignments. Reads were also aligned to the ARS-UCD1.2 assembly.

### Variant calling, postfiltering, and statistics
Variants were called using Freebayes v0.9.21[81] specifying a minimum base quality of 20, a minimum alternate fraction of 0.20, a minimum alternate count of 2, a haplotype length of 0, and a ploidy level of 2. We used a custom python script by setting to missing individual variants whose depth was <1/3 or >2.5 the average genome coverage, as estimated by Qualimap. BCFtools v1.19 was used to further discard SNPs closer than 5 bp to insertions/deletions (InDels), InDels closer than 5 bp to other InDels, variants with a PHRED-quality score <30, and variants with an allele count <2. Finally, BCFtools stats was used to obtain statistics on the final set of called variants. Only autosomal bi-allelic SNPs (InDels excluded) were used in the downstream analyses.

### Genome-wide heterozygosity
Heterozygosity was calculated in 1 Mb sliding windows as the number of heterozygous bi-allelic SNPs divided by the total number of bases that had >1/3 and <2.5-times the average genome coverage[82,83]. Heterozygosity was corrected for the number of sites that were excluded because of coverage. Windows with less than 60% of bases within a normal coverage range were excluded.

### Detecting runs of homozygosity
We identified runs of homozygosity (ROH) using the approach presented in ref. 82, which uses a corrected measure of heterozygosity estimated in 10 Kb windows[83]. The heterozygosity threshold within a candidate ROH was relaxed to allow peaks of heterozygosity if their inclusion did not inflate the heterozygosity within the final ROH, which had to be below 0.25 the average heterozygosity. This minimized the impact of local assembly or alignment errors.

### Realized genomic inbreeding
The realized genomic inbreeding coefficient ($F_{ROH}$) was estimated from the sum of autosomal ROH longer than 100 Kb divided by the genome length of the first 29 autosomes in the wisent genome ($L = 2,682,350,267$ bp).

### Coverage analysis near the *THRSP* deletion
We assessed coverage near the *THRSP* deletion in the six short read-sequenced wisent samples and 719 publicly available short read samples of wisent, bison, taurine, cattle, indicine cattle, and water buffalo aligned to ARS-UCD1.2 with samtools depth with flags "-aa -r 29:17990000-18000000". Accession numbers of the DNA sequencing data are provided in Supplementary Data 3. Coverage was normalized based on the mean sequencing depth across this interval, excluding the deletion region (29:17993500-17996000).

### Transcriptome analyses
Publicly available RNA sequencing data from bison (BioProject: PRJNA257088[37]) and water buffalo (BioProject: PRJNA951806[40]) were aligned to the ARS-UCD1.2 assembly and Refseq version 106 annotation with STAR v2.7.9a[84]. Integrative Genomics Viewer v2.14.0[85] was used to

visualize the alignments. Accession numbers of the RNA sequencing data are listed in Supplementary Data 3. Transcript abundance was quantified using the kallisto v0.46.1 software[86]. Gene expression from *Bos taurus taurus* transcriptomes was obtained from a publicly available TPM matrix (https://zenodo.org/records/7560235) built by the cattle Genotype-Tissue Expression (GTEx) project[39]. Data from the cattle GTEx were filtered by tissue and only samples obtained from females older than 8 months were retained. Gene expression was averaged over these samples and compared to gene expression in the bison sample using Spearman correlation coefficients.

## Reporting summary

Further information on research design is available in the Nature Portfolio Reporting Summary linked to this article.

## Data availability

The primary assembly of the wisent is publicly available in the European Nucleotide Archive (ENA) under accession GCA_963879515.1 (https://www.ebi.ac.uk/ena/browser/view/GCA_963879515.1). The annotation of the primary assembly is currently underway at Ensembl. HiFi reads of the F1 are available in the ENA at the study accession PRJEB71066 under sample accession SAMEA114863253. Illumina paired-end reads of six captive wisents are available in the ENA at the study accession PRJEB71066 under sample accessions SAMEA115388352, SAMEA115388353, SAMEA115388354 (F1), SAMEA115388355 (dam), SAMEA115388356 (sire), SAMEA115388357. The source data behind the graphs in the paper can be found in Supplementary Data 4.

## Code availability

Codes used in this study are available in GitHub (https://github.com/cbortoluzzi/WisentGenomeAssembly) and zenodo (https://zenodo.org/records/14056475)[87].

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

## Acknowledgements

We thank Stefan Hoby from the Tierpark Bern and Martin Kilchenmann from the Tierpark Langenberg for providing DNA and tissue samples used in this study. We are thankful for the technical support provided by Dr. Anna Bratus-Neuenschwander from the ETH Zurich technology platform Functional Genomics Center Zurich (https://fgcz.ch) for sequencing and DNA fragment analysis. This study was supported by the Swiss National Science Foundation (SNSF, grant ID 204654).

## Author contributions

C.B. characterized genomic diversity, identified ROH, analyzed repeats and synteny, built phylogenetic trees, and drafted the paper; X.M.M. prepared DNA for sequencing; S.N. and F.J. sampled tissue; H.P. conceived the study, examined SV diversity, investigated putative trait-associated SVs, performed transcriptome analyses, and drafted the paper; A.S.L. assembled genomes, investigated the impact of ROH on the assembly process, built and decomposed pangenomes, contributed to the analysis of putative trait-associated SVs, and drafted the paper. All authors read and approved the final manuscript.

## Competing interests

The authors declare no competing interests.
