## [Transparent Peer Review file · Communications Biology]

Genome assembly of wisent (*Bison bonasus*) uncovers a deletion that likely inactivates the THRSP gene

Corresponding Author: Professor Hubert Pausch

Version 0:

Reviewer comments:

Reviewer #1

(Remarks to the Author)

The manuscript by Chiara Bortoluzzi and colleagues presents a high-quality genome assembly of the wisent, revealing a significant deletion in the THRSP exon that deactivates the thyroid hormone-responsive gene. This study utilizes PacBio HiFi reads to construct a detailed wisent genome assembly and establishes a bovine super-pangenome that includes assemblies from various cattle species. A notable 1,580 bp deletion, which removes the first exon of the THRSP gene, was identified. This deletion is present in bison but absent in *Bos* and *Bubalus*, likely influencing the low fat content observed in bison milk and meat. The high-quality wisent genome undoubtedly enhances our understanding of bovine genomes. I have several major questions and minor comments that I hope will help improve the quality of the paper.

Major comments:

1. It is a pity that the analysis based on the generated data was too simple. The authors took a lot of words on the data processing and quality assessment meanwhile few on biological significances. For example, the study built a pangenomes but only presented the number of SVs and discussed a single case. However, what is the distribution of these SVs? How many of SVs located in the cis regulatory element?
2. The authors utilized the HiFi sequencing technology to assemble a high-quality genome and performed genotyping using parental second-generation data. However, it is puzzling that the authors did not use Hi-C data to further enhance the assembly quality to the chromosomal level. Section 1 and Section 2 should be merged into one section which comprehensively represents the genome assembly.
3. The legends and figures need to be improved. For instance, the legend and color in Figure 1.f, the legend for the horizontal axis in Figure 3.a, the coordinates in Figure 3.d and the legend for the different colors of the dots in Fig4.a.

Minor comments:

1. I recommend that the authors include an image of the wisent species in Fig. 1. This visual addition would provide readers with a clearer context and enhance their understanding of the species being discussed.
2. In Fig. 1, a comparison between this assembly and previous versions will better demonstrate the improvement in genome assembly quality. A visual presentation could effectively highlight these enhancements and provide a clear, comparative overview.
3. The completeness of the assembled BUSCO genes should not be isolated in a separate column in Table 1.
4. In Figure 3, the authors provided a phylogenetic tree constructed using BUSCO genes, but the confidence intervals and divergence times between species should be provided. Additionally, have the authors validated the stability of the tree topology through other methods, such as constructing a tree based on whole-genome data?
5. Some abbreviations, such as "water buffalo" mentioned in line 243, have been previously defined in Line 240.
6. Line 248-250, I suggest to break this part of the content into several sentences.
7. Line 300-319, I suggest that the authors add two diagrams for clarification.

Reviewer #2

(Remarks to the Author)

This is a manuscript presenting a new high-quality assembly of the wisent genome, and some related analyses. I liked the structure and scope of the manuscript and found the analyses to be convincingly presented. I therefore only have a few minor comments for improvements, and I recommend acceptance pending these small changes.

L178: I don't see this result in Fig. 2b. Looks like an erroneous figure reference. But I do think it's relevant to add a ROH length distribution result as part of Fig. 2, given the amount of emphasis the authors place on discussing this.

L189-190: "Moreover, our findings emphasize that the average genome-wide heterozygosity can be a misleading metric for haplotype-resolved analyses" - it was not clear to me what the authors mean by this statement.

L240-241: This sentence is poor English and needs modification: "...the water buffalo (*Bubalus bubalis*) was the most outgroup species".

L256: Replace "pair-wise" with "pairwise".

L272: I realize that insertion and deletion are relative terms, but given that the authors convincingly show that the ancestral state is to have the 1580bp sequence, I think they should delete the sentence: "while the 1,580 bp insertion (relative to wisent and bison) is the ancestral state", as it is implied by saying that the deletion happened on the Bison branch.

Fig. 4A: How does assembly quality influence the detection of indels? It looks like the species differ widely in the number of unique (and total) indels, which I would assume could be impacted by data artifacts.

Also in Fig. 4A: Can the authors add information in the legend about what the numbers to the left of the upset plot mean? I was expecting them to be the sum of the numbers of indels in the different bars that contain a given species, but that does not seem to be the case?

L300-319: Can the authors show that the apparent lack of expression of THRSP in Bison is exceptional? I mean, that this is not simply an artifact of e.g. poor or incomplete transcriptome data from the Bison. I imagine this could be done by showing that Bison otherwise has normal gene expression for genes that are found to be expressed in other bovine species.

I think the authors could elaborate on the finding that wisent appears to have higher heterozygosity outside of ROHs than cattle, as this has important implications for understanding the long term demographic history of the two species. This suggests that wisent had larger effective population sizes in the more distant past than cattle. It might also be informative to add a PSMC analysis to emphasize this.

Version 1:

Reviewer comments:

Reviewer #1

(Remarks to the Author)

The authors have largely addressed my concerns, and I have no more question.

Reviewer #2

(Remarks to the Author)

I also reviewed the previous version of the manuscript, and I generally found (and find) it a good contribution to the field. The following comments are just minor comments for the authors to consider, not major prerequisites for acceptance.

Something seems to be missing in the rebuttal letter - I deduced that the letter changes from Reviewer #1 to my (Reviewer #2) comments on page 5, but that was not indicated by any text. Also, some parts of my comments regarding the content of L178 has been deleted, but the response looks meaningful enough.

The authors do not remark on the fact that the tree generated from presence/absence of SVs in Fig. 4b does not match the species tree in *Bos* based on genomic sequence data (e.g. Sinding et al. 2021), according to which gaur should be more closely related to bison and yak than to cattle. I think this requires at least a comment and attempt at explanation/reconciliation.

Related to the comment above, the authors' argument that the number of SVs is a function of evolutionary distance to *Bos taurus taurus* seems not to hold, i.e. gaur should have the same number of SVs relative to *B. t. t.* as bison and yaks do. I would like the authors to consider their response in light of this, i.e. are they convinced that the results for gaur are not affected by assembly quality? What are the likely alternative explanations?

I am doubting whether I agree with the authors' argument that a high overall correlation between taurine and bison gene expression proves that the total absence of THRSP expression in bison is real. I would consider it more relevant to show how many genes that are expressed in cattle, but not in bison, including those that are intact/functional (genetically) in both species.

Version 2:

Reviewer comments:

Reviewer #2

(Remarks to the Author)

I am happy with the revision and have no further comments. I congratulate the authors on a nice study and responsive and efficient revision process.

Reviewer #1 (Remarks to the Author):

The manuscript by Chiara Bortoluzzi and colleagues presents a high-quality genome assembly of the wisent, revealing a significant deletion in the *THRSP* exon that deactivates the thyroid hormone-responsive gene. This study utilizes PacBio HiFi reads to construct a detailed wisent genome assembly and establishes a bovine super-pangenome that includes assemblies from various cattle species. A notable 1,580 bp deletion, which removes the first exon of the *THRSP* gene, was identified. This deletion is present in bison but absent in *Bos* and *Bubalus*, likely influencing the low fat content observed in bison milk and meat. The high-quality wisent genome undoubtedly enhances our understanding of bovine genomes. I have several major questions and minor comments that I hope will help improve the quality of the paper.

Major comments:

1. It is a pity that the analysis based on the generated data was too simple. The authors took a lot of words on the data processing and quality assessment meanwhile few on biological significances. For example, the study built a pangenomes but only presented the number of SVs and discussed a single case. However, what is the distribution of these SVs? How many of SVs located in the cis regulatory element?

We are happy to see that the reviewer appreciates our attention to detail when processing, QC'ing and analysing the wisent sequencing data and assemblies. A thorough analysis of the assemblies was required to reveal that the missing BUSCO genes in the haplotype assemblies were often within extended runs of homozygosity. Give their relevance for many ongoing sequencing efforts, we feel that the detailed description of these findings in the main text is warranted.

We show how the novel wisent assembly helps to address biological questions across phylogenies by investigating a bison-specific SV that inactivates *THRSP*. A lack of functional data from wisent samples (e.g, transcriptomic and epigenomic data, (molecular) phenotypes) prevent us from conducting further investigations but the analysis of comprehensive whole-genome sequencing and transcriptomic data from different species provides compelling evidence that the deletion is private to *bison*.

Annotations for cis regulatory elements are suboptimal/incomplete for the species considered here, and such an SV-overlap analysis with these poorly annotated features is unlikely to be useful. However, we do also show that we can recover the expected phylogenetic relationships purely through SVs, demonstrating the genome-wide value of our SV analysis. Moreover, Ensembl is currently annotating our primary assembly, so that an annotated reference-quality assembly of the European bison will be available for the genomics community soon.

We provide the full list of 96 SVs private to the *Bison* clade that were predicted to have a high impact on protein coding genes in Supplementary File 1. Following the reviewer's suggestion, we examined the distribution of SVs that are private to each haplotype along the chromosomes. We observed a strikingly high fraction of SVs private to the wisent assembly on chromosome 7. This is now presented in a supplementary figure.

Newly added text in the manuscript (lines 258-263)

We find a pronounced increase of SVs private to wisent on chromosome 7 between 10 and 10.6 Mb (27 times the genome-wide rate of private wisent SVs). Many of the private SVs were near or overlapping genes in the olfactory receptor 7 subfamily A (OR7A), with a total of 12 annotated protein coding genes in this region (Supplementary Figure 3), suggesting wisent may have unique variants mediating olfaction compared to the other bovids considered here.

Newly added Supplementary Figure:

Sup Figure 4: Excess of private structural variants to wisent on chromosome 7. (a) Chromosome 7 was a substantial outlier when considering the number of SVs private to wisent compared to the total number of SVs per chromosome and was not a pattern observed in any other sample. (b) Most private SVs to wisent were clustered between 10-10.6 Mb on chromosome 7, a region containing 12 annotated protein coding genes. Two of these genes are explicitly identified as part of the olfactory receptor family 7 subfamily A (OR7A), while many of the remaining genes have orthologous relationships to the OR7A subfamily.

2. The authors utilized the HiFi sequencing technology to assemble a high-quality genome and performed genotyping using parental second-generation data. However, it is puzzling that the authors did not use Hi-C data to further enhance the assembly quality to the chromosomal level.

The HiFi sequencing was based on a biobanked sample from a wisent trio provided by a Swiss zoo, where there was insufficient remaining biological material to collect additional sequencing data types like Hi-C or fresh biological material to collect RNA sequencing.

Section 1 and Section 2 should be merged into one section which comprehensively represents the genome assembly.

Given reviewer's 2 comments on the genome-wide heterozygosity and ROH analysis, we decided to keep the two sections separate but expand on the role of ROH and heterozygosity in the wisent genome (line 166-167, 183-184).

3. The legends and figures need to be improved. For instance, the legend and color in Figure 1.f, the legend for the horizontal axis in Figure 3.a, the coordinates in Figure 3.d and the legend for the different colors of the dots in Fig4.a.

We have improved the figures and captions throughout the manuscript.

Minor comments:

1. I recommend that the authors include an image of the wisent species in Fig. 1. This visual addition would provide readers with a clearer context and enhance their understanding of the species being discussed.

We do not have images of any of the new samples discussed in the manuscript, nor do we feel an image of a wisent would meaningfully contribute to a clearer context for the results presented in our manuscript.

2. In Fig. 1, a comparison between this assembly and previous versions will better demonstrate the improvement in genome assembly quality. A visual presentation could effectively highlight these enhancements and provide a clear, comparative overview.

Quality metrics for the draft assembly and the newly generated primary and draft assemblies are presented in Table 1 in the main body of the manuscript. The two text paragraphs corresponding to Table 1 (lines 114 - 134) provide a detailed comparison between the newly generated and the draft assembly. Given the extremely fragmented nature of the draft assembly, a meaningful visual comparison with our highly contiguous assembly is not possible. Neither circos plots nor chromosome ideograms can meaningfully visualise close to 30k scaffolds (or nearly 250k contigs) from the draft assembly side-by-side with 210 scaffolds (or 248 contigs) from our newly generated primary assembly.

3. The completeness of the assembled BUSCO genes should not be isolated in a separate column in Table 1.

We have kept the single and duplicated copy values and removed the complete=single+duplicated row as suggested by the reviewer.

4. In Figure 3, the authors provided a phylogenetic tree constructed using BUSCO genes, but the confidence intervals and divergence times between species should be provided. Additionally, have the authors validated the stability of the tree topology through other methods, such as constructing a tree based on whole-genome data?

As suggested by the reviewer, we have added divergence times to the tree and performed bootstrapping to investigate the stability of the tree topology. The bootstrap percentages were higher than 94% for all branches in the tree demonstrating the BUSCO-based tree is robust. We show in Fig 4b that a tree constructed from pangenome-wide SVs has the same topology (for the *Bos* and *Bison* genera) as the BUSCO-derived tree, again demonstrating the stability of the tree topology as they are created through completely orthogonal methods. Estimating divergence times required adding additional samples with fossil records. The tree in Figure 3c, the corresponding figure caption, and the corresponding text in the main text (lines 225-250) have been updated accordingly. The topology and divergence times we estimated based on the BUSCO genes are also similar to estimates using nuclear whole-genome sequences. This is now mentioned in the revised manuscript.

Newly added text in the manuscript (lines 236-238)

The topology and divergence estimates of the Bovini subset of the tree are in line with a phylogenetic reconstruction from nuclear whole-genome sequences (Wang et al., 2018).

5. Some abbreviations, such as "water buffalo" mentioned in line 243, have been previously defined in Line 240.

We have removed the repeated use of “water buffalo (*Bubalus bubalis*)”.

6. Line 248-250, I suggest to break this part of the content into several sentences.

We have rewritten this section to flow better with several smaller sentences while keeping the original meaning.

Rewritten text (lines 251-257):

*We then built per-autosome super-pangenomes with the five *Bos* and two *Bison* assemblies, excluding *Bubalus bubalis* due to the different assembled karyotype. Although *Bos gaurus* has a Robertsonian translocation between chromosomes 1 and 29 (Mastromonaco et al., 2004), this genome was not assembled through the centromere fusion, leaving 29 separate assembled autosomes and so was included. We assessed the structural variant (SV) diversity (Fig. 4a), finding the wisent sample contains 74,770 SVs (insertions: 37,814, deletions: 36,956) relative to *Bos taurus taurus*, matching previous findings for other distantly related bovids (Crysnanto et al., 2021; Leonard et al., 2023).*

7. Line 300-319, I suggest that the authors add two diagrams for clarification.

We have added a panel to Supplementary Figure 4 that shows the expression of *THRSP* in 55 tissues from the cattle GTEx dataset.

new panel a) in Supplementary Figure S4:

Figure S4a): Expression of *THRSP* in 55 tissues from the cattle GTEx dataset. The bars represent TPM values, and the black dots represent the number of samples per tissue. The y axis is truncated at 1000. Blue colour represents tissues for which transcriptome data are also available for the 3 years old bison cow.

L178: I don't see this result in Fig. 2b. Looks like an erroneous figure reference. But I do think it's

We have added a panel for the distribution of ROH length, now corresponding to Fig 2b, which also fixes the previously erroneous reference.

Reworked Fig 2 and updated caption:

Figure 2. Genome-wide heterozygosity and runs of homozygosity. (a) Total length of runs of homozygosity (ROH) versus genome-wide heterozygosity in each wisent ($n = 8$), American bison ($n = 4$), and taurine cattle ($n = 4$) sample. (b) Histogram of ROH lengths derived from 10 Kb bins for the same samples from (a), with substantially longer ROHs present in wisent. (c) Histogram of heterozygosity in 1 Mb windows for the same samples from (a), showing wisent have both many more low heterozygosity bins but also an increase in heterozygosity in some regions.

L189-190: “Moreover, our findings emphasize that the average genome-wide heterozygosity can be a misleading metric for haplotype-resolved analyses” - it was not clear to me what the authors mean by this statement.

We have rephrased this to clarify that analyses which depend on heterozygosity (e.g., trio binning) may appear appropriate based on genome-wide levels of heterozygosity but fail in many regions which are extremely homozygous.

Rephrased text (lines 187-190):

Moreover, our findings emphasize that the average genome-wide heterozygosity can be a misleading metric to consider when conducting haplotype-resolved analyses, as long stretches of the genome may be homozygous and thus unable to be assigned into haplotypes, even for relatively “normal” genome-wide heterozygosity levels.

L240-241: This sentence is poor English and needs modification: “...the water buffalo (*Bubalus bubalis*) was the most outgroup species”.

Rephrased as “most distantly related species”. (line 243)

L256: Replace “pair-wise” with “pairwise”.

Fixed (line 263, and also on line 288).

L272: I realize that insertion and deletion are relative terms, but given that the authors convincingly show that the ancestral state is to have the 1580bp sequence, I think they should delete the sentence: “while the 1,580 bp insertion (relative to wisent and bison) is the ancestral state”, as it is implied by saying that the deletion happened on the Bison branch.

As pointed out, the meaning is already clear from the first part of the sentence (which has been split into two sentences for better fluency, line 279), and so we have removed the suggested text.

Fig. 4A: How does assembly quality influence the detection of indels? It looks like the species differ widely in the number of unique (and total) indels, which I would assume could be impacted by data artifacts.

Low assembly quality (for older PacBio CLR or ONT r9 assemblies) would primarily affect small indels if there was insufficient polishing with short reads. However, minigraph only

captures larger SVs (≥ 50 bp), which is almost unaffected by assembly quality (as we have shown previously e.g., <https://www.nature.com/articles/s41467-022-30680-2>). The primary cause of the different number of total or unique SVs is due to evolutionary distance/relationships (e.g., *Bos Indicus* is most closely related to *Bos Taurus*, the source of reference coordinates we decomposed the pangenome into, and so has fewer SVs).

Rephrased text (lines 254-257):

We assessed the structural variant (SV) diversity (Fig. 4a), finding the wisent sample contains 74,770 SVs (insertions: 37,814, deletions: 36,956) relative to Bos taurus taurus, matching previous findings for other distantly related bovids (Crysnanto et al., 2021; Leonard et al., 2023).

Also in Fig. 4A: Can the authors add information in the legend about what the numbers to the left of the upset plot mean? I was expecting them to be the sum of the numbers of indels in the different bars that contain a given species, but that does not seem to be the case?

They are the total number of SVs per assembly. Since the intersection values are rounded, the sum may be slightly off. We have added some clarification to this caption.

Updated Figure 4 caption (line 285):

(a) UpSet plot of SVs called from the 29 autosomes, where the total number of SVs per assembly is shown on the left and the number of intersecting SVs shown above each grouping. The pink and green markers highlight SVs private to both yaks or bison.

L300-319: Can the authors show that the apparent lack of expression of THRSP in Bison is exceptional? I mean, that this is not simply an artifact of e.g. poor or incomplete transcriptome data from the Bison. I imagine this could be done by showing that Bison otherwise has normal gene expression for genes that are found to be expressed in other bovine species.

We used data from the cattle GTEx to compare gene expression between wisent and other bovine species and found overall a very high correlation the tissues (muscle and liver) that had sufficient data for both species. We have added this information to the main body of the manuscript and provide more details about the methods in the material and methods section.

newly added text in the main section (lines 318 - 325):

We mapped the bison transcriptomes to the Bos taurus taurus reference sequence and compared gene expression with age- and sex-matched bovine samples from cattle GTEx for liver and muscle, i.e., two tissues with high THRSP expression and a decent number of informative GTEx samples ($n_{liver}=14$; $n_{muscle}=43$). The Spearman correlation coefficient estimated for 17,150 genes was 0.876 and 0.878 for liver and muscle tissue, respectively, indicating that overall gene expression levels in these tissues correlate well between bison

and cattle. However, as expected given the deletion, we did not detect expression of the coding exon of THRSP in any of the bison tissues (Supplementary Fig. 4f).

newly added text in the M&M section (lines 506 - 508):

Data from the cattle GTEx were filtered by tissue and only samples obtained from females older than 8 months were retained. Gene expression was averaged over these samples and compared to gene expression in the bison sample using Spearman correlation coefficients.

I think the authors could elaborate on the finding that wisent appears to have higher heterozygosity outside of ROHs than cattle, as this has important implications for understanding the long term demographic history of the two species. This suggests that wisent had larger effective population sizes in the more distant past than cattle. It might also be informative to add a PSMC analysis to emphasize this.

Such a PSMC analysis was previously examined in e.g. Figure 1 of Gautier et al. 2016, showing that wisent had a larger effective population size than cattle approximately 1,000 to 10,000 years before present, whereas now it is smaller. We have added an additional sentence referencing their analysis and connecting to the regions of increased heterozygosity we observe in the wisent samples.

Added text (lines 183-184):

However, some genomic regions had higher heterozygosity in wisent than in cattle or American bison, reflecting their larger ancient effective population size (Gautier et al., 2016).

Reviewer #1:

The authors have largely addressed my concerns, and I have no more question.

We thank the reviewer for the positive comments regarding our revisions.

Reviewer #2 :

I also reviewed the previous version of the manuscript, and I generally found (and find) it a good contribution to the field. The following comments are just minor comments for the authors to consider, not major prerequisites for acceptance.

We thank the reviewer for the encouraging comments, and have addressed the remaining points.

Something seems to be missing in the rebuttal letter - I deduced that the letter changes from Reviewer #1 to my (Reviewer #2) comments on page 5, but that was not indicated by any text. Also, some parts of my comments regarding the content of L178 has been deleted, but the response looks meaningful enough.

We apologise for the accidental truncation of the reviewer comment and an unclear transition from Reviewer #1's to Reviewer #2's comments. The ROH distribution was indeed included as initially suggested in the original, full length comment.

The authors do not remark on the fact that the tree generated from presence/absence of SVs in Fig. 4b does not match the species tree in Bos based on genomic sequence data (e.g. Sinding et al. 2021), according to which gaur should be more closely related to bison and yak than to cattle. I think this requires at least a comment and attempt at explanation/reconciliation.

We agree. Previous research indicated gaur is more closely related to bison and yak than to cattle. However, our SV-based tree (Figure 4b) agrees with the BUSCO-based tree we show in Figure 3c, which are two completely independent approaches using different sources of data (genes versus structural variants). It is possible the trees differ slightly (given the divergence of gaur to cattle is almost the same as gaur from bison/yak) due to having only a single sample for each species. We have included a new supplementary figure 3 showing the bootstrap values, which had the lowest support (94/100) for the gaur branch out of all branches. We believe these possibilities largely explain the minor differences observed in the placement of gaur. Although speculative, we also note our gaur sample was also sourced from the Omaha Henry Doorly Zoo, which is referenced in Sinding et al. 2021 as “the two *B. gaurus* from Omaha's Henry Doorly Zoo (ID 199911001-2) possess a private divergent haplotype, basal in clade (2) (Figure 1B) to *B. gaurus* and gayal from East India and Southeast Asia”, which potentially indicates our sample may have some history of hybridization with cattle.

New text (line 238-241):

The topology and divergence estimates of the Bovini subset of the tree are generally consistent with a phylogenetic reconstruction from nuclear whole-genome sequences. Gaur has been previously reported to be more closely related to bison and yak than to cattle²⁹⁻³¹, although a slightly lower bootstrap confidence (Supplementary Figure 3) and only using a single representative assembly for each external node may explain this minor discrepancy.

New Supplementary Figure 3

Related to the comment above, the authors' argument that the number of SVs is a function of evolutionary distance to *Bos taurus taurus* seems not to hold, i.e. gaur should have the same number of SVs relative to *B. t. t.* as bison and yaks do. I would like the authors to consider their response in light of this, i.e. are they convinced that the results for gaur are not affected by assembly quality? What are the likely alternative explanations?

We note based on the implied evolutionary distance in Figure 4b that gaur is almost as distant to cattle as bison and yak are. As the reviewer correctly points out "gaur should have the same number of SVs relative to *B. t. t.* as bison and yaks do", which is exactly what we show. In the SV intersection shown in Figure 4a, our gaur assembly has 70k SVs with respect to *B. t. t.*, which is much closer to the observed 74-75k seen in bison and yak compared to the 40k seen for indicine cattle. Indeed, in the upsetplot of Figure 4a, gaur and all bison and yak assemblies share nearly 15k SVs, which is one of the largest intersections.

I am doubting whether I agree with the authors' argument that a high overall correlation between taurine and bison gene expression proves that the total absence of THRSP expression in bison is real. I would consider it more relevant to show how many genes that are expressed in cattle, but not in bison, including those that are intact/functional (genetically) in both species.

We found 19 unique genes that are highly expressed (>20 TPM) in taurine liver and muscle but not or only negligibly expressed (<0.05 TPM) in bison of which only 2 genes were disparately expressed in both tissues. This has been added to the manuscript:

New text (lines 327-330)

There were 13 and 8 genes in liver and muscle, respectively, which were highly expressed (>20 TPM) in cattle, but not expressed ($0 \leq \text{TPM} < 0.05$) in bison. Only two genes, THRSP and MSMP, exhibited disparate expression patterns across both tissues, highlighting the rarity of this phenomenon.

Comprehensive genomic data from 20 wisent and 19 bison samples unanimously support the 1580-bp deletion containing the functional exon 1 of *THRSP*, but we had access to transcriptome data of just one bison cow. While *THRSP* was not expressed at all in the four bison tissues studied (including liver and muscle in which *THRSP* is highly expressed in cattle and buffalo), we agree that data from more bison and wisent samples are required to ultimately prove that the 1580-bp deletion represents a functional knock-out of *THRSP* in *bison*. To better appreciate this uncertainty, we slightly modified the title of our manuscript and two statements in the abstract and discussion to

Old title: *Wisent genome assembly uncovers extended runs of homozygosity and a large deletion that inactivates the thyroid hormone responsive gene*

New title: *Wisent genome assembly uncovers extended runs of homozygosity and a large deletion that **likely** inactivates the thyroid hormone responsive gene*

Line 26:

*“..that the deletion **likely** inactivates THRSP in bison...”*

Line 337:

*“Collectively, these findings **suggest** that the non-coding exon does not produce mRNA in bison which supports that the deletion of the coding first exon inactivates THRSP and that bison and wisent are lacking the thyroid hormone-responsive protein.”*